# Does acute soccer heading cause an increase in plasma S100B? A randomized controlled trial

Megan E. Huibregtse[1☉], Madeleine K. Nowak[1☉], Joseph E. Kim[1], Rachel M. Kalbfell[1], Alekhya Koppineni[1], Keisuke Ejima[2], Keisuke Kawata[1,3]*

**1** Department of Kinesiology, Indiana University School of Public Health-Bloomington, Bloomington, Bloomington, Indiana, United States of America, **2** Department of Epidemiology and Biostatistics, Indiana University School of Public Health-Bloomington, Bloomington, Indiana, United States of America, **3** Program in Neuroscience, College of Arts and Sciences, Indiana University, Bloomington, Indiana, United States of America

☉ These authors contributed equally to this work.
* kkawata@indiana.edu

## Abstract

The purpose of this study was to test the effect of subconcussive head impacts on acute changes in plasma S100B. In this randomized controlled trial, 79 healthy adult soccer players were randomly assigned to either the heading (n = 41) or kicking-control groups (n = 38). The heading group executed 10 headers with soccer balls projected at a speed of 25 mph, whereas the kicking-control group performed 10 kicks. Plasma samples were obtained at pre-, 0h post-, 2h post- and 24h post-intervention and measured for S100B. The primary hypothesis was that there would be a significant group difference (group-by-time interaction) in plasma S100B at 2h post-intervention. Secondary hypotheses included (1) no significant group differences in plasma S100B concentrations at 0h post- and 24h post-intervention; (2) a significant within-group increase in S100B concentrations in the heading group at 2h post-intervention compared to pre-intervention; and (3) no significant within-group changes in plasma S100B in the kicking-control group. Data from 68 subjects were available for analysis (heading n = 37, kicking n = 31). There were no differences in S100B concentrations between heading and kicking groups over time, as evidenced by nonsignificant group-by-time interaction at 2h post-intervention (B = 2.20, 95%CI [-22.22, 26.63], p = 0.86) and at all the other time points (0h post: B = -11.05, 95%CI [-35.37, 13.28], p = 0.38; 24h post: B = 16.11, 95%CI [-8.29, 40.51], p = 0.20). Part of the secondary outcome, the heading group showed elevation in plasma S100B concentrations at 24h post-intervention compared to pre-heading baseline (B = 19.57, 95%CI [3.13, 36.02], p = 0.02), whereas all other within-group comparisons in both remained nonsignificant. The data suggest that 10 bouts of acute controlled soccer headings do not elevate S100B concentrations up to 24-hour post-heading. Further dose-response studies with longer follow-up time points may help determine thresholds of acute soccer heading exposure that are related to astrocyte activation. The protocol was registered under ClinicalTrials.gov (NCT03488381; retrospectively registered.).

**Data Availability Statement:** All relevant data are within the paper and its Supporting Information files.

**Funding:** This work was partly supported from the Indiana Spinal Cord & Brain Injury Research Fund from the Indiana State Department of Health (KK; ISCBIRF 0019939; https://www.in.gov/isdh/23657. htm) and IU School of Public Health faculty research grant program (KK; FRGP: 2246237). The funders had no role in study design, data collection and analysis, decision to publish, or preparation of the manuscript.

**Competing interests:** The authors have declared that no competing interests exist.

# Introduction

Exposure to subconcussive head impacts, or impacts to the cranium that do not result in clinical signs and symptoms of concussions [1, 2], has the potential to lead to long-term neurological consequences, including neurocognitive impairments [3] and chronic traumatic encephalopathy (CTE) [4, 5]. In contact sports such as American football, soccer, ice hockey, and rugby, athletes are prone to experiencing hundreds to thousands of these subconcussive head impacts each season [1, 6, 7]. Particularly, in soccer, frequent subconcussive head impacts occur both intentionally and unintentionally through contact with other players, the ground, and the ball [8]. For example, retrospective questionnaires estimate that a collegiate soccer player performs up to 500 headers during a single season and over 3,000 during the course of a career [9, 10]. Recently, Saunders et al. prospectively collected head impact kinematic data and video-verified all impacts in 28 men's and women's Division III collegiate soccer players across an entire season, finding that approximately 614 headers (ball-to-head contact) occurred per 1000 athlete-exposures in practices and games which comprised the majority of all types of head impacts that occurred during play [11]. It is critical to note that the authors believe the generalizability of this investigation's results are limited by the small sample size and strict criteria used by the research team to verify each impact; thus the reported incidence rate may certainly be an underestimation of header frequency in collegiate soccer players [11]. Furthermore, there is a great need to understand acute and chronic consequences of subconcussive head impact exposure since over 3 million high school and college athletes engage in contact sports in the United States each year [12].

Blood biomarkers have been explored as potential objective diagnostic tools to gauge the severity of brain injury, with some biomarkers are currently incorporated in clinical practice for the detection of brain injury [13–15]. In particular, S100B, a calcium-binding protein enriched in astrocytes, has emerged as blood biomarker for traumatic brain injury (TBI) [16–18]. For instance, elevated S100B is a strong predictor of mortality after sustaining TBI [19] and a recent meta-analysis concluded that acute S100B concentrations (<3h post-injury) are useful in predicting intracranial bleeding in children after concussion with sensitivity and specificity of 97% and 37.5%, respectively [20]. Previous studies have detected elevations in S100B after acute exposure to subconcussive head impacts during practices and games in soccer and American football players [21–24]. However, an opposing line of research indicates that S100B in blood can be elevated not only from the mechanical forces to the brain, but also from exercise and bodily hits [23, 25–27]. For instance, Straume-Næsheim et al. recruited 535 professional soccer players and identified similar levels of S100B elevation after high-intensity exercise, heading drills, and collision during soccer match [28]. In an effort to replicate the data derived from field studies, Dorminy et al. conducted a pilot laboratory study and reported that 5 acute bouts of soccer headings resulted in non-significant elevations in plasma S100B concentrations [29]. However, the former study failed to control for the frequency and magnitude of head impacts and was unable to differentiate the effects of head impacts from exercise, whereas the latter study included 11 total soccer players without a control group. As result, the isolated effect of acute subconcussive head impacts on circulating S100B concentrations over an acute time period has never been rigorously investigated.

Therefore, we conducted a randomized controlled trial to study the time-course response of S100B after acute subconcussive head impacts. Our soccer heading paradigm [30] was used to induce 10 controlled subconcussive head impacts while eliminating extraneous influences that are inherent in field studies, such as bodily hits, fatigue, strenuous exercise, perspiration, and hydration. The heading quantity of 10 was selected to minimize risk to participants while maximizing ecological validity based on head impact frequency reported in field studies of

soccer and American football players [31–34]. The primary outcome was to determine the between-group differences (group-by-time interaction) in change in plasma S100B concentrations at 2h post-intervention after 10 soccer headings. We hypothesized that the heading group will show significantly greater changes in S100B concentrations between pre- and 2h-post intervention compared to the kicking-control group. This primary time point of 2h-post intervention was determined based on the half-life of S100B and previous reports on S100B's diagnostic utility for concussion [20, 22, 35]. We also tested three secondary hypotheses: (1) there would be no significant between-group differences in change in S100B concentrations at 0h post- and 24h post-intervention (2) there would be a significant within-group increase in plasma S100B in the heading group at 2h post-intervention, while within-group changes in plasma S100B in the heading group at 0h post- and 24h-post would be nonsignificant; and (3) there would be no significant within-group changes in plasma S100B concentrations in the kicking-control group at any time point.

## Methods

### Trial design and randomization

This single-blind, randomized controlled clinical trial examined the changes in plasma S100B in response to an acute bout of ten soccer headers. Participants were randomly assigned to either the soccer heading or kicking-control group using a simple, dice-based randomization method. Subjects were unblinded to their assigned group, but biomarker experimenters were blinded from the group assignment information. Plasma samples were collected at four time points: pre-, 0h post-, 2h post-, and 24h post-intervention. Between the pre and 0h post-intervention time points, participants in the heading group performed ten soccer headers (see Soccer heading intervention section below), and participants in the kicking-control group kicked the soccer ball. Between the 0h post- and 2h post-intervention time points, participants remained in the laboratory and were instructed to refrain from strenuous physical or cognitive activities. Participants returned to the laboratory approximately 24h after the intervention for the final time point. The Indiana University Institutional Review Board (IU IRB) approved the study, and study procedures were performed in accordance with regulations of the IU IRB (protocol registered under ClinicalTrials.gov: NCT03488381). Written informed consent was obtained from all participants.

The trial protocol was registered 7 months after the commencement of the study. This late registration was due to the authors' misunderstanding of the use of the soccer heading protocol being categorized as an interventional trial until one of federal agencies suggested otherwise. The trial registration occurred on April 5, 2018, and the first participant was enrolled on August 31, 2017. Thirty-seven participants were enrolled in the study prior to the registration, which accounts for 55% of the final sample size. No interval analysis was conducted prior to the registration. The authors confirm that all ongoing and related trials for this intervention are registered.

### Participants

From August 2017 until May 2019, using a convenient sampling strategy, we recruited potential participants who were enrolled at Indiana University—Bloomington, met the following inclusion criteria, and were free of exclusion criteria. Inclusion criteria included being between 18 and 26 years old and having at least five years of soccer heading experience, which ensures their proficiency to perform soccer headings [35, 36]. Exclusion criteria included a history of head injury within 12 months prior to data collection, a history of vestibular, ocular, or visual dysfunction, a history of neurological disorders, or a clinical diagnosis of a learning disability.

Our sample size calculation, based on results from previous studies [21, 37] and a minimal clinical important difference of 25 pg/mL, suggested a total of 56 participants (28 participants per intervention) to yield a statistical power of at least 0.80 with a level of significance of = 0.05. We estimated a worst-case dropout rate of 25%. As a result, a total of 79 participants were recruited in the study and were randomly assigned into the heading (n = 41) and kicking-control (n = 38) groups. Participants were instructed to refrain from any activity that involved head impacts during the study period. At timepoints pre- and 24h post-intervention, participants verified that they had not participated in any head impact activities 24 hours prior to the 24h post-intervention timepoint.

## Soccer heading intervention

A standardized soccer heading intervention was used to induce ten subconcussive head impacts in the form of soccer heading [30, 38]. Bevilacqua et al. [38] contains the video version of the soccer heading intervention. A triaxial accelerometer (SIM-G, Triax Technologies, Inc., Norwalk, CT) was held in place at the occipital protuberance with a custom headband to quantify the linear and rotational acceleration of each head impact. A JUGS soccer machine (JPS Sports, Tualatin, OR) projected a size 5 soccer ball, reaching the participant at a speed of about 25 mph (11.2 m/s). The ball speed is on the slower-scale end of rising balls kicked by adult soccer players [39]. An average linear head acceleration from a header ranges between 26 and 32 $g$ [2], while regular corner or goal kicks (~50mph) yield accelerations above 50 $g$ [40]. This study agreed with our previous studies [2, 41] in that 10 headings did not increase concussion-related symptoms in study participants, ensuring that our intervention is in fact "subconcussive." All participants stood approximately 40 ft (12.2 m) in front of the JUGS machine. Participants in the heading group were instructed to head the soccer ball with their forehead and aim the ball towards a researcher standing approximately 16 ft (4.9 m) in front of the participant. For the heading group, the JUGS machine was set at an angle of 40 degrees from the horizontal by elevating it four inches off the ground. Participants in the kicking-control group were given the same set of instructions, except to kick the ball towards the researcher instead of heading. Participants performed 10 headers or kicks with one-minute intervals between each header or kick. Ten headers was chosen based on previous studies that show soccer players head the ball on average 6–12 times per game [42]. Furthermore, a collegiate American football players incurs on average of 7 to 10 hits per practice, with a mean peak linear acceleration per impact ranging from 28 to 32 g [21, 43]. Therefore, our subconcussive intervention that consists of 10 headers with 33 g per header is translatable beyond soccer.

## Plasma sampling and S100B measurement

At each time point, four milliliters of venous blood were collected into EDTA vacutainer tubes (BD Biosciences, San Jose, CA). Plasma was separated by centrifugation (1500 x $g$, 15 min, 4˚C) and stored at -80˚C until analysis. Plasma S100B concentrations were measured using an enzyme-linked immunosorbent assay (ELISA) kit (Human S100B ELISA, EMD Millipore Corporation, Billerica, MA). The lower detection limit of the assay is 2.7 pg/mL using a 50 μL plasma sample size, and the assay covers a concentration range of up to 2000 pg/mL, with an inter-assay variation of 1.9–4.4% and an intra-assay variation of 2.9–4.8%. Samples were loaded in duplicate into the ELISA plates according to manufacturer instructions. Fluorescence was measured by a microplate reader (BioTek EL800, Winooski, VT) and converted into pg/mL as per the standard curve concentrations. To eliminate the inter-assay effect on within-subject data, all samples from each participant were assayed on the same plate. The biomarker experimenters were blinded from the group assignment information.

Past literature has estimated that S100B is cleared rapidly following mTBI, with the half-life of circulating S100B to be approximately 60 to 120 minutes [17, 22, 44–46]. For example, in a study examining mTBI patients, Townend et al. estimated the elimination half-life of S100B from circulation after mTBI to be 97 minutes [44]. This estimated half-life is well within the duration of our acute timepoints used in this study (0h post-, 2h post-, and 24h post-intervention).

### Primary and secondary outcomes

The primary outcome was the between-group difference in change in plasma S100B at 2h post-intervention. The secondary outcomes were: (1) between-group differences in change in S100B concentration at 0h post- and 24h post-intervention (group-by-time interaction); (2) within-group changes (time effects) in plasma S100B concentrations in the heading group; and (3) within-group changes (time effects) in plasma S100B concentration in the kicking-control group.

### Statistical analysis

Demographic differences between the heading and kicking-control groups were assessed using Mann-Whitney U tests for not normally distributed continuous variables (age, BMI, number of previous concussions, years of soccer heading experience) or Fisher's exact test for a categorical variable (sex). The effect of soccer heading on acute plasma S100B concentrations was assessed using a mixed effects regression model (MRM). A MRM was constructed to regress time, intervention, and time by intervention interaction on the plasma S100B concentrations. The model was adjusted for BMI. Participants were treated as a random effect to account for individual S100B differences at the baseline. Cook's distance was calculated to examine unusual influence of individual data points on the fit of the model. Two data points were identified as outliers, such that plasma S100B concentrations were found to be greater than 700 pg/mL, well above high levels observed in previous S100B studies of subconcussive head impacts [22, 33, 34, 47, 48] and TBI [49–51], and the model was refit without these two points. All Mann-Whitney U tests were two-tailed, and the significance level was set a priori to 0.05. Any measurements below the detection limit of the assay were treated as missing data points. The analysis approach was intention-to-treat (ITT). Missing data points were treated as Missing Completely At Random (MCAR), and thus missing values were not imputed. All analyses were conducted using R (version 3.4.1) with packages "lmer" and "lmerTest."

## Results

### Demographics and head impact kinematics

Eighty-four individuals were evaluated for eligibility. Seventy-nine participants, who met inclusion criteria and were free of exclusion criteria, proceeded to randomization. There were 11 voluntary withdrawals (heading n = 4, 22.5 (1.0) years old, 0% male; kicking-control n = 7, 20.9 (1.1) years old, 57% male) prior to the pre-intervention time points. Data from 68 participants (n = 37 heading, n = 31 kicking-control) were available for analysis (Fig 1). Demographics and head impact kinematics are presented in Table 1. There were no significant differences in any of the demographic variables between groups.

### Primary outcome: Between-group difference in change in S100B at 2h post-intervention

Ten acute soccer headings did not result in group difference in change in S100B concentrations at 2h post intervention, as evidence by non-significant group-by-time interaction (B = 2.20, 95%CI [-22.22, 26.63], p = 0.86; Fig 2).

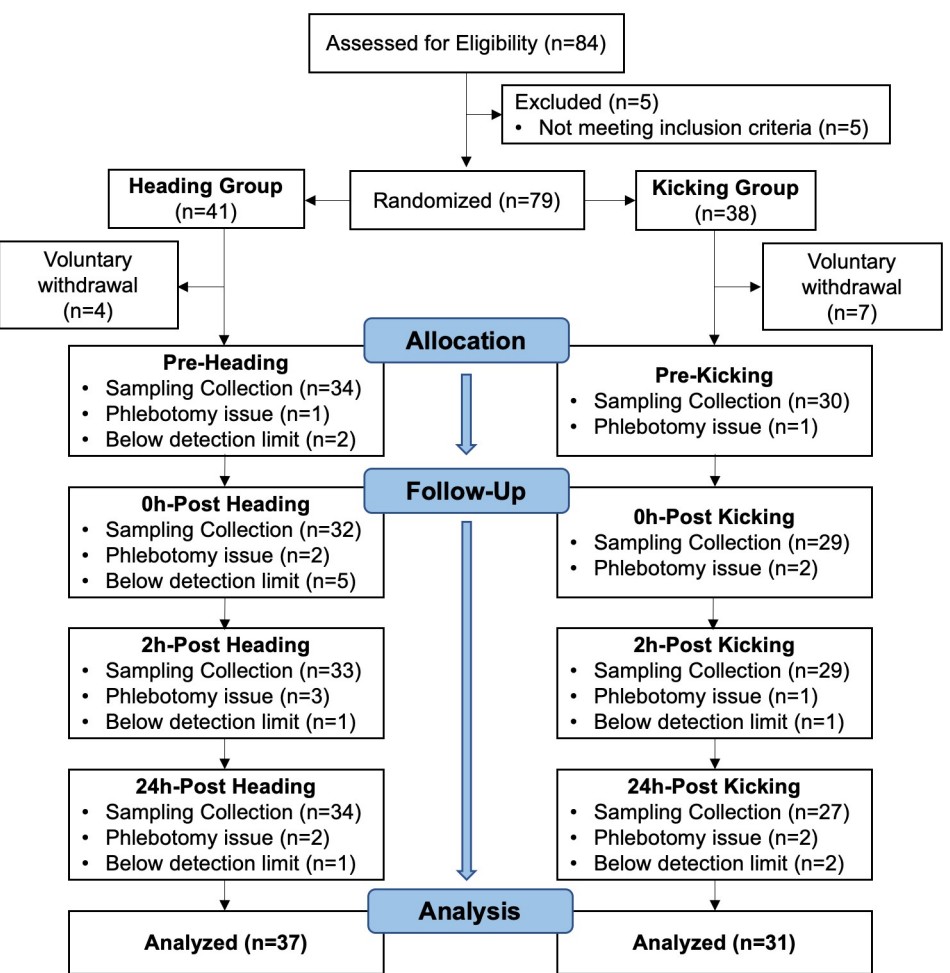

**Fig 1. Study flow chart of eligibility assessment, randomization, data collection, and analysis.**

**Table 1. Demographics and impact kinematics by group.**

| Variables | Heading Group | Kicking-control group | P-value |
|---|---|---|---|
| *Demographics* | | | |
| n | 37 | 31 | - |
| Sex | 19M 18F | 14M 17F | 0.635 |
| Age, y | 21 (19–22) | 21 (20–22) | 0.294 |
| BMI, kg/m$^2$ | 23.4 (21.5–25.2) | 23.4 (22.5–25.8) | 0.389 |
| No. of previous concussion | 0 (0–1) | 0 (0–0) | 0.215 |
| Soccer heading experience, y | 9 (6–11) | 9 (6–12.8) | 0.599 |
| *Head impact kinematics, mean ± SD* | | | |
| PLA, *g* | 33.2 ± 6.8 | - [a] | - |
| PRA, krad/s$^2$ | 3.6 ± 1.4 | - [a] | - |

Note: All data are presented as median (interquartile range), unless otherwise specified. BMI, body mass index. PLA, peak linear acceleration. PRA, peak rotational acceleration. krad, kiloradian.

[a]Soccer kicking did not cause a detectable level of head acceleration.

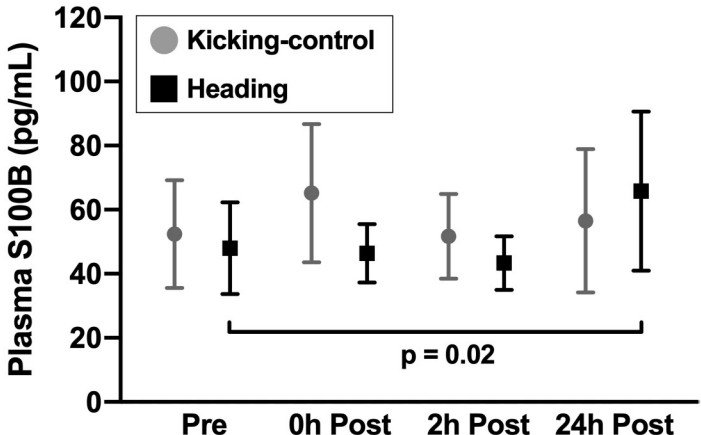

**Fig 2. Plasma S100B concentrations in the heading and kicking-control groups at each study time point (pre-, 0h post-, 2h post-, and 24h post-intervention).** Data are presented as means with the error bars representing the 95% confidence intervals. There was a significant within-group increase in plasma S100B in the heading group at 24h post-intervention relative to pre-intervention (p = 0.02). There were no other significant time, group, or group-by-time effects of soccer heading on plasma S100B concentrations.

## Secondary outcome: Between-group differences in change in S100B at 0h and 24h post-intervention and within-group change over time

There was no significant group-by-time interaction in change in S100B concentrations at 0h or 24h post-intervention (0h post: B = -11.05, 95%CI [-35.37, 13.28], p = 0.38; 24h post: B = 16.11, 95%CI [-8.29, 40.51], p = 0.20; Fig 2). There was a significant within-group elevation in plasma S100B at 24h post-intervention in the heading group (B = 19.57, 95%CI [3.13, 36.02], p = 0.02). There were no other significant time effects for both groups (see Table 2).

## Discussion

To our knowledge, this is the first randomized controlled trial to examine an acute time-course expression of plasma S100B after subconcussive head impacts across three post-head-impact time points (0h, 2h, and 24h). Although the implication of the data is limited to the acute post-impact phase, we provide evidence to suggest that 10 controlled soccer headers are not

**Table 2. Within-group time effects.**

|  | Estimate (B) | 95%CI | P-value |
|---|---|---|---|
| *Heading group* |  |  |  |
| 0h post-intervention | -0.27 | [-17.06, 16.52] | 0.98 |
| 2h post-intervention | 0.99 | [-15.94, 17.92] | 0.91 |
| 24h post-intervention | 19.57 | [3.13, 36.02] | 0.02* |
| *Kicking-control group* |  |  |  |
| 0h post-intervention | 10.78 | [-6.83, 28.39] | 0.23 |
| 2h post-intervention | -1.22 | [-18.82, 16.38] | 0.89 |
| 24h post-intervention | 3.47 | [-14.56, 21.49] | 0.71 |

Note: Within group changes are in reference to S100B concentrations at pre-intervention baseline.

* p < 0.05,

** p < 0.01, *** p < 0.001.

sufficient to provoke astrocyte activation, as reflected by a lack of significant change in S100B concentration in the heading group relative to the kicking-control group.

Although we failed to distinguish S100B elevation in soccer headings from a kicking control, S100B has been demonstrated to significantly correlate with severity, recovery outcomes (e.g., mortality, disability), and cerebrovascular and neuronal cellular damage from TBI and subconcussive head impacts [19, 52–54]. In a cohort of 92 patients with TBI admitted within 12 hours of injury, Pelinka et al. detected a relationship between S100B and mortality rates, as evidenced by significantly higher S100B concentrations in non-survivors compared to survivors at all six time points, ranging from admittance to 108h post-injury [19]. The authors also found a significant positive relationship between S100B concentrations and the severity of intracranial bleeding through CT scans [19, 55]. These findings were consistent with Ingebrigtsen et al., who detected a significant association between elevated S100B concentrations and abnormal neuroradiological findings, such as cranial fractures and brain contusions (CT and MRI) [56]. The same group examined S100B concentrations in 278 patients with TBI and found that higher S100B concentrations were associated with worse injury severity as determined by lower Glasgow Coma Scale scores. Bazarian et al. found that white matter alterations were associated with both concentrations of S100B autoantibodies and head impact kinematic variables in a cohort of collegiate football players over the course of one season [52]. Despite the increasing literature, thresholds identifying what type and magnitude of head impacts elicit concussion and subconcussion remain unclear. Concussion symptom provocation can be influenced by additional factors such as age, sex, location of impact, and recovery periods [1]. The lack of a definite threshold underscores the need for objective measures, such as blood biomarkers, to capture the consequences of subconcussive head impacts.

Despite the clinical utility of S100B as a biomarker of brain injury, it has long been in debate that S100B can also be translocated to the bloodstream from several extracranial cellular sources, such as Schwann cells, ganglion cells, adipocytes, and skeletal myofibers [57, 58]. Aside from neurotrauma, additional variables, such as exercise [25–27, 59], race [60], mood disorder diagnosis [61], and alcohol consumption [62], have been shown to have an influence on plasma S100B concentrations. Physical exertion has been shown to result in acute increases in serum S100B concentrations, pointing to the difficulty of accurate interpretation when athletes incur head trauma and exhibit elevations in serum S100B [25–27, 59]. Dietrich et al. recruited 16 elite swimmers and observed a significant elevation in serum S100B concentrations from pre- (70.7±17.7 pg/mL) to post-competition (108.1±19.5 pg/mL) [25]. Furthermore, acute elevations in S100B concentrations have been detected across a wide range of running intensities and durations [26, 27]. However, Kiechle et al. were able to distinguish the proportional increase in serum S100B concentration after sport-related concussions from sport-related non-contact exertion levels in young adult athletes (AUC 0.904), suggesting that TBI-induced elevations of S100B are far beyond those of physical exercise effects.

Our motivation to conduct the current randomized controlled trial was to validate the growing number of studies supporting the use of circulating S100B concentrations to examine the effects of subconcussive head impacts [21–23, 29, 33, 34, 47, 63]. Acute increases in serum S100B concentration have been detected from pre- to post-game in both male and female soccer players, and the magnitude of S100B increase has been correlated with the number of headers that each player performed in the game [33, 34]. For example, studies by Kawata et al., Zonner et al., and Marchi et al. reported that serum S100B concentrations increase after high school and collegiate American football games and practices by nearly 300% compared to pre-game/practice baseline, with the degree of S100B elevation correlating to the number and magnitude of head impact sustained during game/practice [21, 22, 24]. However, despite our previous effort to control for muscle damage through creatine-kinase levels and exercise effects

through excess post-exercise oxygen consumption (EPOC) [22], "true" head impact effects on circulating S100B concentrations can only be tested in a controlled environment, such as our soccer heading intervention. Clinical studies have reported that the magnitude of soccer headings by collegiate female soccer players can reach as high as 71.2 $g$ (90th percentile head impact of any type by PLA) [64], and male soccer players perform as many as 19 headers per game [34]. Using the Head Impact Telemetry System, Duma et al. identified that the average PLA of head impacts during American football practices and games is 32 $g$, with nearly 90% of all head impacts falling below 60 $g$ [43]. Collegiate American football players incur frequent head impacts during practices and games in an intensity-dependent manner: shell-only practice, avg. 12.7 impacts/player; full gear practice, avg. 16.8 impacts/player; game, avg. 25 impacts/player, with an similar head impact magnitude of 28 $g$ across any type of practice and game [32, 65]. Therefore, our soccer heading model, which was comprised of 10 soccer headers with an average PLA of 33.2 $g$, is representative of a bout of subconcussive head impacts sustained in contact sports while also minimizing risk to participants and maximizing feasibility and ecological validity.

In addition to the present study, previous laboratory studies, using soccer heading models in an attempt to isolate subconcussive head impacts from confounding variables such as exercise, have been unable to replicate the findings of clinical studies. Namely, Dorminy et al. did not detect significant changes in serum S100B concentrations in 11 college-aged soccer players following a bout of 5 headers, despite investigating three different ball speeds (30, 40, and 50 mph) [29]. These results corroborated the findings of Stålnacke et al., who found that S100B concentrations did not differ from baseline at 0.5h, 2h, or 4h post-heading [47]. Furthermore, the serum S100B concentrations in the heading group did not differ from a non-heading control group at all time points. In agreement with the aforementioned studies, the current study did not detect significant between-group differences in changes in S100B concentrations following an acute bout of soccer heading, despite increasing the sample size, including a kicking-control group, and extending the study time frame to 24h post-intervention. The within-group elevation in plasma S100B in the heading group at 24h post-intervention merits some discussion, despite the lack of between-group difference at the same timepoint. The 24h post-intervention timepoint lies outside the half-life of S100B in the blood, suggesting that either the effect of 10 soccer headers may take considerably longer than anticipated to result in a peripheral increase in S100B or there were behaviors or factors that we did not account for between the 2h and 24h post-intervention timepoints driving this increase withing the heading group. The prolonged time for expression and translocation to the bloodstream could suggest that the mechanism by which S100B from astrocyte activation increases in the periphery is predominantly through clearance by the glymphatic system rather than blood-brain barrier (BBB) disruption [17]. The presumable low level of neural damage from a short bout of soccer heading may not be sufficient to disrupt the BBB as seen with more severe head injuries [66, 67]. The glymphatic system, which is more active during sleep [68], could be the predominant avenue by which S100B is cleared from the interstitial space through paravenous space after 10 subconcussive head impacts in the form of controlled soccer headers, explaining why we observed an increase when participants returned the next day for the 24h post-intervention blood sample collection. Furthermore, astrocyte activation has been previously noted to continue for up to 20 hours post trauma [69]. Therefore, the timeline of observed S100B elevation could surpass the half-life. The other possible explanation, albeit less likely, for the within-group increase in plasma S100B at 24h post-intervention in the heading group may be that before returning for the last timepoint, some or many participants within the heading group partook in activities that have been shown to result in elevations in peripheral S100B, such as swimming, running, or weight training [23, 25, 26]. We feel that this is unlikely due to the fact

that participants were randomized to one of the two conditions, reducing the likelihood of this potential explanation for a late increase in plasma S100B in just the heading group. Again, it is important to note that we did not detect a between-group difference in change in plasma S100B at this timepoint, thus we can cannot conclude that soccer heading resulted in a meaningful increase in plasma S100B. Future research should explore reasons for this delayed elevation in S100B.

## Clinical implication

The use of blood biomarkers in the diagnosis of brain injury is emerging due in part to its objectivity. The results of the present study contribute to the body of head injury literature that should be considered when monitoring athletes and making clinical decisions. When investigating possible neural damage in response to subconcussive head impacts, the use of S100B should not be relied upon alone. Especially in low head impact exposure such as ~10 headings, S100B may not be useful to surrogate neurologic stress. Nonetheless, it is important that practitioners pair S100B with additional objective methods, such as neuroimaging or other blood biomarkers (e.g., neurofilament light [NF-L], tau, glial fibrillary acidic protein [GFAP]) [70], in addition to subjective symptom reporting from the patient. Furthermore, emerging data on salivary-based S100B, which has shown to adequately differentiate concussion patients from controls (AUC = 0.74) [71], may have a direct implication to clinical practice in near future. Nonetheless, multimodal validation of biomarker findings is needed.

## Limitations

The results of the present study should be interpreted in light of several limitations. We did not monitor participants' behavior or activity between 2h and 24h post-intervention time points, although participants were instructed to avoid situations where they might sustain subconcussive head impacts. There is a possibility that factors about which we were unaware outside the study protocol may have contributed to the response of S100B concentration. Other potential confounding variables, such as malignant melanoma [72], major depression disorder [73], sleep quality and quantity, diet, or menstrual cycle phase, were not accounted for; however, our repeated measures study design with participant randomization and a linear mixed effects regression model should minimize any potential subject-level influences on plasma S100B concentrations. We acknowledge that there are other assay methodologies for the detection of S100B such as the Elecsys and Cobas systems (Roche Diagnostics). However, our ELISA method has been utilized in a number of previous studies including Dorminy et al. [29] and our studies [21, 37], and the majority of S100B concentrations in the present study were well within the threshold for CT referral. Our study used a standardized frequency of 10 soccer headings, thus our data can only suggest that 10 headings or fewer do not significantly elevate S100B concentrations. As we do not know the consequence of headings beyond 10 hits, it should not be generalized or concluded that acute subconcussive head impacts are safe. Lastly, we encountered a 15% dropout rate, which was mostly attributed to changes in participants' schedules during the semester or loss of interest, given that all participants were full-time college students. The present study was conducted in a single-site setting, which limits generalizability of the finding.

## Conclusion

There is growing concern that even mild head impacts can cause significant insidious neurotrauma if sustained repetitively. S100B has been suggested to be effective in identifying concussion and more severe forms of TBI; however, subtle changes caused by subconcussive head

impacts are more challenging to detect. Our data suggest that plasma S100B is not sensitive enough to monitor acute exposure to subconcussive head impacts from 10 controlled soccer headers. Future studies should continue to investigate S100B and other promising blood biomarkers for subconcussive head impact exposure.

## Supporting information

**S1 Data.**
(CSV)

**S1 File.**
(DOCX)

**S2 File. CONSORT 2010 checklist of information to include when reporting a randomised trial***.
(DOCX)

## Acknowledgments

We would like to thank Zachary W. Bevilacqua for his assistance with data collection and Angela M. Wirsching for her help with study administration.

## Author Contributions

**Conceptualization:** Keisuke Kawata.

**Data curation:** Megan E. Huibregtse, Madeleine K. Nowak, Keisuke Kawata.

**Formal analysis:** Megan E. Huibregtse, Madeleine K. Nowak, Keisuke Ejima.

**Funding acquisition:** Keisuke Kawata.

**Investigation:** Megan E. Huibregtse, Madeleine K. Nowak, Joseph E. Kim, Rachel M. Kalbfell, Alekhya Koppineni.

**Methodology:** Keisuke Kawata.

**Project administration:** Keisuke Kawata.

**Resources:** Keisuke Kawata.

**Supervision:** Megan E. Huibregtse, Madeleine K. Nowak, Keisuke Ejima, Keisuke Kawata.

**Visualization:** Megan E. Huibregtse, Madeleine K. Nowak, Keisuke Ejima, Keisuke Kawata.

**Writing – original draft:** Megan E. Huibregtse, Madeleine K. Nowak, Rachel M. Kalbfell, Alekhya Koppineni.

**Writing – review & editing:** Megan E. Huibregtse, Madeleine K. Nowak, Joseph E. Kim, Rachel M. Kalbfell, Alekhya Koppineni, Keisuke Ejima, Keisuke Kawata.

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
