## [Decision Letter · Decision Letter 0]

30 Jun 2020

PONE-D-20-15329

Does acute soccer heading cause an increase in plasma S100B? A randomized, controlled clinical trial

PLOS ONE

Dear Dr. Kawata,

Thank you for submitting your manuscript to PLOS ONE. After careful consideration, we feel that it has merit but does not fully meet PLOS ONE’s publication criteria as it currently stands. Therefore, we invite you to submit a revised version of the manuscript that addresses the points raised during the review process.

While I find the study well-written and organized, some major concerns were raised by the reviewers and by me. Please, consider replying carefully all questions.

Abstract, line 27: which hypothesis was tested? (please state it).

Abstract, purpose: please consider modifying your purpose focusing on testing effect rather than examining relationship.

Abstract, results: Please add effect sizes and not just p-values. Focus on group contrasts in the reporting (please see the PREPARE Trial guide for guidance).

Abstract, conclusion: The study hypothesis was superiority, which was not supported. So, the authors need concluding  that clearly. Please conclude on the primary outcome firstly and secondary outcomes secondly and separately. Please make sure that the conclusion in the abstract is identical to the one I the manuscript text, if revised.  

Abstract, end: Please add clinical-trial registration-info at the end of the abstract. Because it seems as if the trial was retrospectively registered (registration after inclusion of the first participant) add “retrospectively registered” after the trial registration number. Please state clearly in the manuscript if the primary outcome was pre-defined (defined before inclusion of the first participant).

Introduction: To make sure placing your research in context, please include level 1a evidence (systematic reviews) if possible (for example systematic reviews on biomarkers related to concussion/traumatic brain injury).

Outcomes: Outcomes are the variables tested/compared, please consider changing that, indicating which variables were the primary and secondary outcomes. Analyzing your register a clinical trials site I wonder why you did not include Ocular-Motor Function (Over Time in Relation to the Baseline) measures? Please, include all analysis as registered. And the blood biomarkers? Just S100B?

Hypotheses

Consider including more objective hypotheses, not just diff or not, but y higher than x condition style.

Registration: Please check if any differences exist between that registered in Clinical.Trials.gov and that reported in the manuscript. This includes: outcomes, objective, in- and exclusion criteria etc. Please explain any changes.

Inclusion: Which sampling strategy was used? Random sampling or convenience sampling, for example? Please state this.

Methods, sample size estimation: Please write out the sample size estimation so that others can replicate it. Please see http://www.bmj.com/content/338/bmj.b1732for guidance. Please state what was considered the minimal clinical important difference (between-conditions).

Was it possible to account for confounding factors across groups, such as the effect of pharmacological doses? 

Statistics: Please pay careful attention to the review from Reviewer 1, who is a statistician. Please state clearly if your analysis approach was intention-to-treat (ITT) and how missing values were imputated. And, most importantly your statistical model needs to be changed to GEE or mixed models, where the ITT should be possible.

Please remove statistical tests for baseline differences. CONSORT advise against this. Please see http://www.consort-statement.org/Media/Default/Downloads/CONSORT%202010%20Explanation%20and%20Elaboration%20Document-BMJ.pdf  page 17.

Table 1, there is line without data. Consider adjusting the table removing vertical lines and marking in italic the subtitles (kinematics).

Results and Stats: please report 95CI of all variables and effect sizes.

Discussion: One para addressing some potential applications of your findings can be useful for practitioners

Discussion: given that S100b is a blood Marker of intense Brain injury, consider discussing on potential mechanisms explaining the similarities.

We look forward to receiving your revised manuscript.

Kind regards,

Leonardo A. Peyré-Tartaruga, Ph.D.

Academic Editor

PLOS ONE

Journal Requirements:

2. Thank you for submitting your clinical trial to PLOS ONE and for providing the name of the registry and the registration number. The information in the registry entry suggests that your trial was registered after patient recruitment began. PLOS ONE strongly encourages authors to register all trials before recruiting the first participant in a study.

a) your reasons for your delay in registering this study (after enrolment of participants started);

b) confirmation that all related trials are registered by stating: “The authors confirm that all ongoing and related trials for this drug/intervention are registered”.

Please also ensure you report the date at which the ethics committee approved the study as well as the complete date range for patient recruitment and follow-up in the Methods section of your manuscript.

Reviewers' comments:

Reviewer's Responses to Questions

**Comments to the Author**

1. Is the manuscript technically sound, and do the data support the conclusions?

Reviewer #1: Partly

Reviewer #2: Yes

2. Has the statistical analysis been performed appropriately and rigorously? 

Reviewer #1: No

Reviewer #2: Yes

3. Have the authors made all data underlying the findings in their manuscript fully available?

Reviewer #1: No

Reviewer #2: Yes

4. Is the manuscript presented in an intelligible fashion and written in standard English?

Reviewer #1: Yes

Reviewer #2: Yes

5. Review Comments to the Author

Reviewer #1: The authors here have looked at a randomised trial of some 68 participants proceeding to intervention. The use of randomisation is to be applauded here - although one does question whether a crossover design might have helped control for heterogeneity between subjects better.

The reason for the sample size and the size of difference deemed important here is not given. This is crucial to understanding the minimum important difference to be looked at in S100b. Indeed actual differences (b as opposed to the normalised beta coefficients) and their confidence intervals are required to be given and inference based upon the confidence intervals - if no difference is seen can one rule out a meaningful difference or is this absence of evidence as opposed to evidence of absence. This is crucial here.

Some variables are clearly not Normal (eg where the sd is more than half the mean) so t-tests are not appropriate, and indeed the data cannot properly be represented suing mean and sd in tables.

In terms of secondary outcome measures what are the results of a repeated measures analysis?

In the CONSORT diagram, reasons for dropout are required. It seems strange to be happy to commit to either group and then drop out before the actual intervention.

Reviewer #2: Review original research article

“Does acute soccer heading cause an increase in plasma S100B? A randomized, controlled clinical trial”

This randomized controlled clinical study aimed to examine the relationship between subconcussive head impacts and changes in plasma S100B. S100B is a blood biomarker that has been explored as potential objective diagnostic tool to gauge the severity of brain injury. In this context, previous studies have detected elevations in S100B after acute exposure to subconcussive head impacts. Therefore, this study examined the acute response of S100B after heading multiple times. As S100B is a strong predictor of mortality after sustaining TBI, it is worth examining if heading in soccer might have an impact on S100B values. The manuscript is well written and the structure is clear and comprehensible. Therefore, the structure and language/grammar etc. does not require much revision. However, there are some aspects that need revision in order to be considered being published.

Major comments:

Currently, it is unclear whether heading effects players’ brain anatomy and physiology. Therefore, the scientific debate continues about a potential header-induced brain damage for elite and amateur football players. Acute effects of heading on blood biomarkers such as S100B are important in the light of possible short- or long-term consequences. However, in the present study 10 headers performed by football players were chosen. Some important questions remain:

What was the rationale to choose 10 headers? Are 10 headers supposed to simulate the heading incidence in football to investigate the acute effects of heading? Are 10 headers a mean number of headers that are performed in real-life by players? I would like to see this clearly mentioned in the introduction and methods section. As this is an artificial laboratory experiment, I think it is necessary to explain how this experiment may be linked to realistic match play with heading exposure. Please specify.

Minor comments:

Introduction page 2 lines 44-46: The spectrum of long-term consequences of traumatic brain injuries or subconcussive head impacts either caused by a single or by repetitive head traumas, mainly refers to three severe medical conditions as they represent the most clinically relevant and severe examples: chronic post-concussion syndrome (PCS), neurocognitive impairments (e.g. mild cognitive impairment (MCI) up to the point of dementia), and the chronic traumatic encephalopathy (CTE). Maybe add the first two conditions as potential long-term consequences.

Introduction page 3 lines 50-52: These studies used questionnaires to assess heading numbers. Such numbers should be interpreted with caution. Please add prospective data of real-life heading in soccer.

Introduction page 3 lines 72-73: Here, heading is defined as a sub-concussive blow. The term “sub-concussive” describes a cranial impact with potential neuronal changes similar to those in concussion, but without the symptoms of a concussion (Bailes et al. 2013). I’d like to see this aspect sufficiently discussed in the discussion section, in detail is there a certain threshold to be a sub-concussive event etc.?

Methods page 5 line 109: Any information on sample size calculations?

Methods page 5 lines 110-111: Participants were instructed to refrain from any activity that involved head impacts during the study period, but did you control for head impacts prior to the investigation (e.g. the day before etc.)?

Methods page 5 lines 115-116: Here you cite ref 27, in the introduction ref 24, both appear to be the same reference. Please correct.

Methods page 6 line 120: A short information why 25 mph was chosen exactly, although this information might be found in the cited paradigm.

Methods page 6 lines 129 ff: Could you tell the reader something about the half-life of S100B as mentioned on page 4 line 80?

Discussion page 11 lines 222-224: References?

Discussion page 11 lines 228-230: Is it possible to add how far beyond those of physical exercise effects?

Discussion page 12 lines 244-248: This information is important (see my previous comments) to strengthen the purpose of this study. Such info should be added in the introduction and methods section to offer the reader an explanation why exactly 10 headers were chosen. Additionally, what kind of head impacts were differentiated in these studies?

Discussion page 13 lines 264-265: Could you add some examples, which factors might have influenced S100B concentrations despite the ones you already mentioned throughout the manuscript, if any?

Discussion page 13 lines 265-267: Do these (confounding) variables have an influence on S100B concentrations?

References: Please check ref 24 and ref 27.

References page 17 line 371: Delete.

Comment figure 2: Possibly renew this figure. Area 0-100 bigger?

6. PLOS authors have the option to publish the peer review history of their article (what does this mean?). If published, this will include your full peer review and any attached files.

Reviewer #1: No

Reviewer #2: No

---

## [Author Response · Author response to Decision Letter 0]

29 Jul 2020

Dear Dr. Peyré-Tartaruga and Reviewers, 

Thank you very much for your feedback and suggestions. We appreciate the time and effort that went into the review of our manuscript, and we believe that our manuscript has been significantly improved through this revision. We have responded to your points one by one, as follows.

Editorial Comments:

1) “Abstract, line 27: which hypothesis was tested? (please state it).”

RESPONSE: We have revised the abstract to clearly state the primary hypothesis that there would be a significant between-group difference (group-by-time interaction) in plasma S100B at 2h post-intervention. This primary hypothesis (outcome) was followed by a series of secondary hypotheses (outcomes). 

2) “Abstract, purpose: please consider modifying your purpose focusing on testing effect rather than examining relationship.”

RESPONSE: We have revised the first sentence of the abstract in a way that we focused on testing the effect of subconcussive head impacts on acute changes in plasma S100B levels. Thank you for the suggestion. 

3) “Abstract, results: Please add effect sizes and not just p-values. Focus on group contrasts in the reporting (please see the PREPARE Trial guide for guidance).”

RESPONSE: The results in the abstract have been revised to contain the unstandardized �, 95%CI, and p-values. Further, the abstract has been revised to emphasize group contracts.

4) “Abstract, conclusion: The study hypothesis was superiority, which was not supported. So, the authors need concluding that clearly. Please conclude on the primary outcome firstly and secondary outcomes secondly and separately. Please make sure that the conclusion in the abstract is identical to the one in the manuscript text, if revised.” 

RESPONSE: We have revised the abstract to clarify both the hypotheses and the conclusions, which correspond to the main text. 

5) “Abstract, end: Please add clinical-trial registration-info at the end of the abstract. Because it seems as if the trial was retrospectively registered (registration after inclusion of the first participant) add “retrospectively registered” after the trial registration number. Please state clearly in the manuscript if the primary outcome was pre-defined (defined before inclusion of the first participant).”

RESPONSE: The clinical trial registration information has been added to the end of the abstract: ClinicalTrials.gov (NCT03488381; retrospectively registered). As per the IRB protocol (available at the ClinicalTrials registration page), the primary outcomes were established prior to enrollment of the first participant; however, as discussed in our response to comments 7 and 9 below, S100B was added as an additional biomarker of interest after the study had begun. Our rationale for the primary hypothesis hinged on what is known about the half-life of S100B in the bloodstream, which has been added to the “Plasma sampling and S100B measurement” subsection of the Methods. 

6) “Introduction: To make sure placing your research in context, please include level 1a evidence (systematic reviews) if possible (for example systematic reviews on biomarkers related to concussion/traumatic brain injury).”

RESPONSE: We included biomarker systematic review by O’Connell et al. and a meta-analysis on S100B in concussion by Oris et al.

7) “Outcomes: Outcomes are the variables tested/compared, please consider changing that, indicating which variables were the primary and secondary outcomes. Analyzing your register a clinical trials site I wonder why you did not include Ocular-Motor Function (Over Time in Relation to the Baseline) measures? Please, include all analysis as registered. And the blood biomarkers? Just S100B?”

RESPONSE: We have updated the outcomes subsection of the Methods to clarify what was tested for each hypothesis. The trial was designed to address various aspects of neurologic response against 10 repetitive headings. We agree that including all variables in a paper is one way of presenting the data, while we are aware that each modality and biomarker reflect different aspects of neurologic health. For example, ocular-motor testing assesses functional response, whereas S100B is reflective of astrocyte activation. The source of S100B is different from tau or neurofilament (axons). Given that S100B is one of the most extensively studied biomarkers with split results, we therefore focused on S100B in this paper to demonstrate the utility of S100B in gauging subconcussive astrocyte response. 

8) “Hypotheses. Consider including more objective hypotheses, not just diff or not, but y higher than x condition style.”

RESPONSE: Thank you for this suggestion. We had long discussion about this objective hypothesis. Although it may be more captivating and possibly stronger to have objective cut-off levels in our hypothesis (e.g., the heading group will show 60 pg/mL higher levels of S100B at 2h post compared to that of the kicking-control group). However, we ran into an issue of unclear threshold for S100B that is linked to brain damage/astrocyte activation. Previous studies have shown varied degrees of S100B elevation after subconcussive head impacts. This variability is possibly due to testing methods (ELISA vs. Chemiluminescent assay) and uncontrolled extraneous factors in field studies. To this end, we decided to keep our hypothesis style as is but included more direction to our primary and secondary hypotheses.

9) “Registration: Please check if any differences exist between that registered in Clinical.Trials.gov and that reported in the manuscript. This includes: outcomes, objective, in- and exclusion criteria etc. Please explain any changes.

Inclusion: Which sampling strategy was used? Random sampling or convenience sampling, for example? Please state this.”

RESPONSE: There are no differences in any procedural items between the manuscript and registered protocol regarding inclusion/exclusion criteria, outcomes, time points, yet as you pointed out that we did not specify sampling strategy in the manuscript nor the protocol. This is a single-site trial with convenient sampling of young adults who are enrolled as students at Indiana University. We state this in the Participants section. 

10) “Methods, sample size estimation: Please write out the sample size estimation so that others can replicate it. Please see http://www.bmj.com/content/338/bmj.b1732for guidance. Please state what was considered the minimal clinical important difference (between-conditions).”

RESPONSE: We have added the sample size estimation and estimated attrition rate for the primary outcome to the “Participants” subsection of the Methods. The minimal clinical important difference was considered to be a group difference of 25 pg/mL. 

11) “Was it possible to account for confounding factors across groups, such as the effect of pharmacological doses?” 

RESPONSE: This is an important point. The novelty of this study is the use of repeated measures design with controlled intervention that can eliminate many extraneous factors. Recruited subjects are from generally a healthy cohort (college-aged young adults) and we did not observe any group differences in the demographic variables presented in Table 1. Therefore, we decided that the only factor that we would adjust the model by would be BMI, since S100B is known to be released by adipocytes [1, 2]. 

12) “Statistics: Please pay careful attention to the review from Reviewer 1, who is a statistician. Please state clearly if your analysis approach was intention-to-treat (ITT) and how missing values were imputated. And, most importantly your statistical model needs to be changed to GEE or mixed models, where the ITT should be possible.”

RESPONSE: Thank for you for drawing our attention to this ambiguity. We have switched our analysis to use a mixed effect regression model, and missing values were not imputed and were assumed missing completely at random (MCAR). Thus, our analysis approach is still ITT as all available data points are used in the analysis. We have revised the “Statistical Analysis” subsection of the Methods and the Results accordingly.

13) “Please remove statistical tests for baseline differences. CONSORT advise against this. Please see http://www.consort-statement.org/Media/Default/Downloads/CONSORT%202010%20Explanation%20and%20Elaboration%20Document-BMJ.pdf page 17.”

RESPONSE: This has been removed. Thank you for providing this BMJ document. 

14) “Table 1, there is line without data. Consider adjusting the table removing vertical lines and marking in italic the subtitles (kinematics).”

RESPONSE: We have reformatted Table 1 as per your suggestions.

15) “Results and Stats: please report 95CI of all variables and effect sizes.”

RESPONSE: SEs have been replaced with the 95CI for all variables. The non-standardized beta coefficients are given in the text.

16) “Discussion: One paragraph addressing some potential applications of your findings can be useful for practitioners.”

RESPONSE: We have added an additional paragraph (Clinical Implication section, before the limitation section) addressing potential clinical implication of our result, by mostly calling for a caution when using S100B to manage acute subconcussive head impacts. As discussed in this new paragraph, we emphasized the importance of using a panel of biomarker or multimodal approach (e.g., biomarker x neuroimaging) to aid in diagnosis, athlete monitoring, and optimal guidelines for return to play.

17) “Discussion: given that S100b is a blood marker of intense brain injury, consider discussing on potential mechanisms explaining the similarities.”

RESPONSE: The similarities in potential mechanism between mild-moderate-severe TBI and subconcussive head impacts are elaborated throughout the discussion, yet mechanistic insight of S100B as a biomarker for brain injury requires a multimodal approach (perhaps with advanced neuroimaging). We also encourage the multimodal approach as part of clinical implication and limitation sections. Thank you for the suggestion. 

Reviewer Comments

Reviewer #1: 

1) “The reason for the sample size and the size of difference deemed important here is not given. This is crucial to understanding the minimum important difference to be looked at in S100b. Indeed actual differences (b as opposed to the normalised beta coefficients) and their confidence intervals are required to be given and inference based upon the confidence intervals - if no difference is seen can one rule out a meaningful difference or is this absence of evidence as opposed to evidence of absence. This is crucial here.”

RESPONSE: This is a great suggestion. We have added the sample size calculation and the MCID to the “Participants” subsection of the Methods. The regression coefficients given in the manuscript are not normalized (condition and timepoint are dummy-coded; BMI was standardized) and therefore represent the change in S100B (in pg/mL). We have given the confidence intervals, rather than reporting the standard errors, for the coefficients. 

2) “Some variables are clearly not Normal (eg where the sd is more than half the mean) so t-tests are not appropriate, and indeed the data cannot properly be represented suing mean and sd in tables.”

RESPONSE: Thank you for pointing out this oversight. After confirming that these variables did not follow a normal distribution, we have used Mann-Whitney U test and Fisher’s exact test for our demographic variables. Table 1 and the Statistical Analysis section are adjusted accordingly. 

3) “In terms of secondary outcome measures what are the results of a repeated measures analysis?”

RESPONSE: Both the new mixed-effect regression model (MRM) and the original linear quantile mixed model (LQMM) analyses accounted for repeated measures analyses. As suggested by the editor, in this revision, we used MRM, instead of LQMM, to test both primary and secondary outcomes. The results of the secondary outcomes— both the time and group*time at 0h and 24h effects—are reported in the text. In addition, we have moved Supplemental Table 1 (containing the time effects within each group) into the main text (becoming Table 2), which will better inform readers regarding our secondary outcomes. 

4) “In the CONSORT diagram, reasons for dropout are required. It seems strange to be happy to commit to either group and then drop out before the actual intervention.”

RESPONSE: A total of 11 participants (almost entirely undergraduate students) consented to participate in the study and were therefore randomized to an intervention group but voluntarily withdrew during the scheduling process due to a variety of personal reasons, such as loss of interest, running out of free time before exams or an academic break, unpredictable schedule changes, etc. Some subjects provided us with reasons of withdrawal while others stopped responding to our reminders. Therefore, we are unable to provide exact reasons for all 11 withdrawal. To this reason, we kept “voluntary withdrawal” as is and elaborated briefly in the limitation section. 

Reviewer #2: 

Major comments:

1) “What was the rationale to choose 10 headers? Are 10 headers supposed to simulate the heading incidence in football to investigate the acute effects of heading? Are 10 headers a mean number of headers that are performed in real-life by players? I would like to see this clearly mentioned in the introduction and methods section. As this is an artificial laboratory experiment, I think it is necessary to explain how this experiment may be linked to realistic match play with heading exposure. Please specify.”

RESPONSE: This is an important point to clarify here and in the manuscript. The frequency of 10 soccer headers in our model is a perfect balance between feasibility, safety, and clinical implication. If our intervention includes, for instance, 20 headers each day for 4-5 days in a row, it is indeed mimicking some athletes with very high frequent exposure. However, that is unsafe (from IRB standpoint) and infeasible (difficulty in retaining subjects). Heading frequency is largely dependent on playing style, position, levels, and coaching philosophy, but a player heads the ball an average of 6-12 times per game [3]. Furthermore, a collegiate American football players incurs an average of 7 to 10 hits per practice, with a mean peak linear acceleration per impact ranging from 28 to 32 g [4, 5]. Therefore, our subconcussive intervention that consists of 10 headers with 33 g per header is translatable beyond soccer. We have added this explanation to the methods section (“Soccer heading intervention” subsection), and the clinical implication of our intervention was detailed in the 4th paragraph of the discussion section.

Minor comments:

2) “Introduction page 2 lines 44-46: The spectrum of long-term consequences of traumatic brain injuries or subconcussive head impacts either caused by a single or by repetitive head traumas, mainly refers to three severe medical conditions as they represent the most clinically relevant and severe examples: chronic post-concussion syndrome (PCS), neurocognitive impairments (e.g. mild cognitive impairment (MCI) up to the point of dementia), and the chronic traumatic encephalopathy (CTE). Maybe add the first two conditions as potential long-term consequences.”

RESPONSE: We agree that it is better to provide the spectrum of neurologic consequences due to subconcussive head impacts. I addition to CTE, we have added “neurological impairments” to better represent the spectrum of potential consequences of exposure to subconcussive head impacts. We discussed among the team about PCS and given that our focus is subconcussive head impacts, which do not directly lead to PCS; therefore, we omit inclusion of PCS in the introduction. Thank you for the pointing this out. 

3) “Introduction page 3 lines 50-52: These studies used questionnaires to assess heading numbers. Such numbers should be interpreted with caution. Please add prospective data of real-life heading in soccer.”

RESPONSE: Real-life soccer heading frequency data by Saunders et al. (2020) has been added to the introduction to balance the heading data collected from retrospective questionnaires used in Lipton et al. (2013) and Matser et al. (1999).

4) “Introduction page 3 lines 72-73: Here, heading is defined as a sub-concussive blow. The term “sub-concussive” describes a cranial impact with potential neuronal changes similar to those in concussion, but without the symptoms of a concussion (Bailes et al. 2013). I’d like to see this aspect sufficiently discussed in the discussion section, in detail is there a certain threshold to be a sub-concussive event etc.?”

RESPONSE: This is a great point with unclear consensus among the neurotrauma research community. Currently there is no definite threshold identifying what type and magnitude of head impacts elicit symptoms (concussion) and remain asymptomatic (subconcussion). For this reason, some investigators, mainly led by Drs. Talavage and Nauman at Purdue University, lately began using the term “head acceleration event” instead of “subconcussive head impacts”. At the end of the 2nd paragraph in the discussion, we have added further information of how every impact is dynamic and individualized, as well as the difficulties it leads to when tracking neurological deficits. This new addition also provides readers with examples of factors that may influence outcomes from subconcussive impacts.

5) “Methods page 5 line 109: Any information on sample size calculations?

RESPONSE: The sample size calculation has been added to the “Participants” subsection of the Methods.” 

6) “Methods page 5 lines 110-111: Participants were instructed to refrain from any activity that involved head impacts during the study period, but did you control for head impacts prior to the investigation (e.g. the day before etc.)?”

RESPONSE: Additional information on participants responsibility has been added. Participants were asked at the beginning of both days 1 and 2 of the study regarding any activity that involved head impacts. However, we did not control for head impacts outside of this time frame of 24 hours prior to the start of the study. The participants were not in season; thus, minimizing the chances of individuals experiencing head impacts outside of our study. 

7) “Methods page 5 lines 115-116: Here you cite ref 27, in the introduction ref 24, both appear to be the same reference. Please correct.”

RESPONSE: Thank you for bringing this to our attention. Citations and bibliography have been updated accordingly.

8) “Methods page 6 line 120: A short information why 25 mph was chosen exactly, although this information might be found in the cited paradigm.”

RESPONSE: Further detail is provided to justify the chosen characteristics of the soccer heading protocol. The information added allows readers to compare our lab protocol with real world situations.

9) “Methods page 6 lines 129 ff: Could you tell the reader something about the half-life of S100B as mentioned on page 4 line 80?”

RESPONSE: Additional past literature examining S100B half-life in reference to mTBI has been added to the methods section. We also elaborated in the discussion regarding the time-course utility of S100B in identifying the presence of intracranial bleeding in concussion patients. This will provide the reader with current suggested information on the half-life of S100B and the relation to our study protocol.

10) “Discussion page 11 lines 222-224: References?”

RESPONSE: References have been clarified and added as per your suggestion.

11) “Discussion page 11 lines 228-230: Is it possible to add how far beyond those of physical exercise effects?”

RESPONSE: As reported in Kiechle et al. (2014, PLOS ONE), the post-exertional serum S100B levels (mean 0.071±0.03 µg/L; not significantly different from pre-exertion) were not directly compared to 3h post-sports related concussion (SRC) serum S100B levels (mean 0.099±0.008 µg/L; significantly different from baseline: 0.058±0.006 µg/L). However, ROC analyses revealed an AUC of 0.772 for the absolute post-exertion/SRC value and 0.904 for the proportional increase in serum S100B, suggesting that serum S100B can distinguish between non-contact exertion and SRC. We have added the AUC for the proportional increase to illustrate this. 

12) “Discussion page 12 lines 244-248: This information is important (see my previous comments) to strengthen the purpose of this study. Such info should be added in the introduction and methods section to offer the reader an explanation why exactly 10 headers were chosen. Additionally, what kind of head impacts were differentiated in these studies?”

RESPONSE: Agreed, elaborating on the rationale behind selecting 10 headers strengthens the purpose of our study. We have added a detailed explanation to the “Soccer heading intervention” subsection of the methods (as specified above in response to your first comment.) We have also added a brief explanation for why 10 headers were chosen to the last paragraph of the introduction, just before the hypotheses. 

As for your last question, our citation for the 90th percentile peak linear acceleration of head impacts in female collegiate soccer players was McCuen et al. (2015) in which the head impacts were not differentiated by type and included impacts from headers, falls, collisions with other soccer players, and whiplash-like events. Stälnacke et al. (2004) had two independent researchers watch video recordings of games and separately classify acceleration/deceleration events into headers, jumps, falls, and collisions; the range of headers per player per game was 0-19. Duma et al. (2005) and Crisco et al. (2010) differentiate head impacts by location using HITS data, but not by impact type, in their respective studies of collegiate American football player. Kawata et al. (2016) and Reynolds et al. (2016) did not report data on head impact location but did specify collegiate football practice type (pads-on, pads-off, helmet only, etc). Finally, Broglio et al. (2011) reported the cumulative head impact burden in high school football players, differentiated head impacts that occurred in practices from games, and used impact location in the calculation of the HIT severity profile (but did not report any data on the distribution of head impacts by helmet location. We have revised this portion of the discussion to clarify and provide additional detail to justify the selection of 10 headers for the soccer heading model.

13) “Discussion page 13 lines 264-265: Could you add some examples, which factors might have influenced S100B concentrations despite the ones you already mentioned throughout the manuscript, if any?”

RESPONSE: In addition to what was already mentioned throughout the manuscript, studies examining possible influential factors of S100B have now been included into the manuscript. With your suggestion, we provide examples of other potential factors, such as BMI, race, alcohol consumption, and mood disorders with appropriate citations in the 3rd paragraph of the Discussion section.

14) “Discussion page 13 lines 265-267: Do these (confounding) variables have an influence on S100B concentrations?”

RESPONSE: Same as our response above #13, several factors have shown to influence S100B levels. Our controlled heading model was able to eliminate all these factors, which is the novelty of this RCT. 

15) “References: Please check ref 24 and ref 27.”

RESPONSE: Citations and bibliography have been updated accordingly.

16) “References page 17 line 371: Delete.”

RESPONSE: This line is continued from the line above it. It contains the end of the reference above. We have updated the reference and left the format as is.

17) “Comment figure 2: Possibly renew this figure. Area 0-100 bigger?”

RESPONSE: Figure 2 has been reworked to enlarge the 0-100 pg/mL range and to make the image easier to understand. Following our changes to the original analysis, Figure 2 has been updated to reflect our updated analysis of mixed-effect regression model (MRM). This model resulted in a figure that is more readily interpreted.

References

1. Anderson RE, Hansson LO, Nilsson O, Dijlai-Merzoug R, Settergren G. High serum S100B levels for trauma patients without head injuries. Neurosurgery. 2001;48(6):1255-8; discussion 8-60. Epub 2001/06/01. doi: 10.1097/00006123-200106000-00012. PubMed PMID: 11383727.

2. Savola O, Pyhtinen J, Leino TK, Siitonen S, Niemela O, Hillbom M. Effects of head and extracranial injuries on serum protein S100B levels in trauma patients. J Trauma. 2004;56(6):1229-34; discussion 34. Epub 2004/06/24. doi: 10.1097/01.ta.0000096644.08735.72. PubMed PMID: 15211130.

3. Spiotta AM, Bartsch AJ, Benzel EC. Heading in soccer: dangerous play? Neurosurgery. 2012;70(1):1-11; discussion Epub 2011/08/04. doi: 10.1227/NEU.0b013e31823021b2. PubMed PMID: 21811187.

4. Duma SM, Manoogian SJ, Bussone WR, Brolinson PG, Goforth MW, Donnenwerth JJ, et al. Analysis of real-time head accelerations in collegiate football players. Clin J Sport Med. 2005;15(1):3-8. Epub 2005/01/18. PubMed PMID: 15654184.

5. Kawata K, Rubin LH, Lee JH, Sim T, Takahagi M, Szwanki V, et al. Association of Football Subconcussive Head Impacts With Ocular Near Point of Convergence. JAMA Ophthalmol. 2016;134(7):763-9. Epub 2016/06/04. doi: 10.1001/jamaophthalmol.2016.1085. PubMed PMID: 27257799.

---

## [Decision Letter · Decision Letter 1]

9 Sep 2020

Does acute soccer heading cause an increase in plasma S100B? A randomized controlled trial

PONE-D-20-15329R1

Dear Dr. Kawata,

We’re pleased to inform you that your manuscript has been judged scientifically suitable for publication and will be formally accepted for publication once it meets all outstanding technical requirements.

Kind regards,

Leonardo A. Peyré-Tartaruga, Ph.D.

Academic Editor

PLOS ONE

Additional Editor Comments (optional):

Reviewers' comments:

Reviewer's Responses to Questions

**Comments to the Author**

1. If the authors have adequately addressed your comments raised in a previous round of review and you feel that this manuscript is now acceptable for publication, you may indicate that here to bypass the “Comments to the Author” section, enter your conflict of interest statement in the “Confidential to Editor” section, and submit your "Accept" recommendation.

Reviewer #1: All comments have been addressed

Reviewer #2: All comments have been addressed

2. Is the manuscript technically sound, and do the data support the conclusions?

Reviewer #1: (No Response)

Reviewer #2: Yes

3. Has the statistical analysis been performed appropriately and rigorously? 

Reviewer #1: (No Response)

Reviewer #2: Yes

4. Have the authors made all data underlying the findings in their manuscript fully available?

Reviewer #1: (No Response)

Reviewer #2: Yes

5. Is the manuscript presented in an intelligible fashion and written in standard English?

Reviewer #1: (No Response)

Reviewer #2: Yes

6. Review Comments to the Author

Reviewer #1: (No Response)

Reviewer #2: (No Response)

7. PLOS authors have the option to publish the peer review history of their article (what does this mean?). If published, this will include your full peer review and any attached files.

Reviewer #1: No

Reviewer #2: No

---

## [Editor Report · Acceptance letter]

14 Oct 2020

PONE-D-20-15329R1 

Does acute soccer heading cause an increase in plasma S100B? A randomized controlled trial 

Dear Dr. Kawata:

I'm pleased to inform you that your manuscript has been deemed suitable for publication in PLOS ONE. Congratulations! Your manuscript is now with our production department. 

Kind regards, 

on behalf of

Professor Leonardo A. Peyré-Tartaruga 

Academic Editor

PLOS ONE